# Learning Neural Surrogate Model for Warm-Starting Bayesian Optimization

## Abstract

Bayesian optimization is an effective tool to optimize black-box functions and popular for hyper-parameter tuning in machine learning. Traditional Bayesian optimization methods are based on Gaussian process (GP), relying on a GP-based surrogate model for sampling points of the function of interest. In this work, we consider transferring knowledge from related problems to target problem by learning an initial surrogate model for warm-starting Bayesian optimization. We propose a neural network-based surrogate model to estimate the function mean value in GP. Then we design a novel weighted Reptile algorithm with sampling strategy to learn an initial surrogate model from meta train set. The initial surrogate model is learned to be able to well adapt to new tasks. Extensive experiments show that this warm-starting technique enables us to find better minimizer or hyper-parameters than traditional GP and previous warm-starting methods.

## 1 Introduction

Bayesian optimization (Mockus, 1975; Pelikan et al., 1999; Gustavo Malkomes, 2018) is an optimization method to find global minimizer of a black-box function without knowing its convexity, differentiability or continuity. Bayesian optimization iteratively samples new point by an acquisition function, and uses Gaussian process (Rasmussen, 2003; Gardner et al., 2014) to estimate a surrogate model to fit $f(x)$ based on the observed points. When using specific acquisition function, its regret bound has been proved as $O(\sqrt{n})$ ($n$ is the number of sampled points) in Srinivas et al. (2012). Freitas et al. (2012) proposed a method that uses Bayesian optimization with dividing region, which successfully accelerates convergence rate. More algorithms have been developed in Wang et al. (2014); Kawaguchi et al. (2015); Springenberg et al. (2016). In application, Bayesian optimization has been widely applied in machine learning for hyper-parameter tuning (Snoek et al., 2012).

A machine learning problem is formulated as follows. Given data set $\mathcal{D}$, model $\mathcal{M}(\cdot, \lambda)$ and loss function $\mathcal{L}(\cdot, \cdot)$, it aims to find an optimal parameter vector $w$ to minimize $\mathcal{L}(\mathcal{M}(w; \lambda), \mathcal{D})$. In practice, model $\mathcal{M}(w; \lambda)$ is dependent on *hyper-parameter* $\lambda$, which may significantly influence the model. Thus choosing appropriate $\lambda$ is a critical procedure of learning a model. For instance, $\mathcal{M}(w; \lambda)$ is a neural network to solve image classification problem, then $\lambda$ can be the number of hidden units or network layers. Hyper-parameter optimization problem can be formulated as

$$\min_{\lambda, w} \mathcal{L}(\mathcal{M}(w; \lambda), \mathcal{D}) = \min_{\lambda} \mathcal{L}(\mathcal{M}(w^*; \lambda), \mathcal{D}), \text{where } w^* = \mathcal{A}(\mathcal{L}(\mathcal{M}(w; \lambda), \mathcal{D})), \quad (1)$$

and $\mathcal{A}$ is an algorithm minimizing loss function w.r.t. $w$. It is challenging to optimize Eq. (1) since $\mathcal{A}(\mathcal{L}(\mathcal{M}(w; \lambda), \mathcal{D}))$ is a complicated function w.r.t $\lambda$, which might be non-differentiable. In real applications, hyper-parameter is commonly tuned by hand. Random search and grid search are typical methods to find plausible $\lambda$, but they are time-consuming to find the optimal hyper-parameter because they sample the hyper-parameter in a brute-force way.

Bayesian optimization is an effective tool to minimize function for global optimization or hyper-parameter tuning. Bayesian optimization iteratively samples new point and fits the function values by a surrogate model based on Gaussian process. It starts with several random points without using any prior knowledge on the task. In applications, we often solve several related machine learning problems, therefore, one question is that whether we can learn common knowledge from related problems and transfer them to new problem as an initialization for adaptation, commonly referred to as *warm-starting Bayesian optimization*.

Meta-learning is a promising approach for learning optimizer for fast optimization or initial model for fast adaptation to novel task using a set of training problems as meta train set (Andrychowicz et al., 2016; Li & Malik, 2017; Wang et al., 2019; Ravi & Larochelle, 2017; Chen et al., 2017; Finn et al., 2017; Nichol et al., 2018). This idea was also applied to Bayesian optimization in past years. Poloczek et al. (2016) built a joint statistical model of the collection of related objective functions. Feurer et al. (2015) initialized Bayesian optimization based on hand-crafted meta features for hyper-parameter tuning. Given a new problem, its hyper-parameters are initialized by the most similar problem in meta train set with distance of hand-crafted meta features. However, it is challenging to define meta features artificially. Kim et al. (2017) learned meta features of problems by deep metric learning. Wistuba et al. (2016); Feurer et al. (2018) proposed to transfer surrogate model by linear combination of surrogate models of problems in meta train set with coefficients determined by ranking of function values or meta features. However, representation capacity of linear combination is limited, meaning that if surrogate model of a new problem cannot be represented by linear combination of surrogate models of meta train problems, then this approach will fail.

In this work, we propose a novel meta-learning approach to learn prior knowledge from meta train set as a warm-start of unseen related target problem. More specifically, we make the following two contributions to tackle the challenges in traditional Bayesian optimization.

*Firstly*, as we know, traditional Bayesian optimization is data-driven and depends on a GP-based surrogate model fitting function values without learnable parameters, limiting its ability for adaptation to new problems. We propose to use neural network to model the surrogate model (dubbed as neural surrogate model) to substitute the mean of function values in GP. We further design a ranking loss for fitting surrogate model to sampled points to make surrogate model insensitive to the scale of function values.

*Secondly*, we propose a novel meta-learning algorithm to learn an initial neural surrogate model from meta train set of related problems, which is taken as initial surrogate model for adaptation to new problem. To train the initial neural surrogate model, meta-learning methods such as MAML (Finn et al., 2017) are not applicable because Bayesian optimization is non-differentiable. Thus we propose a weighted Reptile algorithm inspired by Reptile algorithm (Nichol et al., 2018) using sampling strategy for exploration to train the initial neural surrogate model.

We evaluate our method on tasks including optimization of synthetic functions, hyper-parameter tuning in evolutionary algorithm, neural network and SVM. These experiments show that our method outperforms traditional Gaussian process and related warm-starting methods in these tasks.

## 2 PRELIMINARIES

### 2.1 BAYESIAN OPTIMIZATION

Bayesian optimization is to find a minimizer $x^*$ of a function $f$ in a bounded space by iteratively querying the function at sampled points. At $n$-th iteration, we first construct a probabilistic model on $D = \{x_i, f(x_i)\}_{i=1}^n$ for estimating function values, then use an acquisition function $a(x)$ to select next point. We estimate $f$ by surrogate model based on Gaussian process (GP). We assume $f(\cdot) \sim GP(m(\cdot), k(\cdot, \cdot))$, where $m(\cdot)$ is mean function and $k(\cdot, \cdot)$ is covariance function or kernel. Given $D = \{x_i, f(x_i)\}_{i=1}^n$, GP is simply a joint Gaussian $\mathbf{f}(x_{1:n}) \sim \mathcal{N}(\mathbf{m}(x_{1:n}), \mathbf{K}(x_{1:n}))$, where $\mathbf{K}(x_{1:n})$ is a $n \times n$ covariance matrix with elements of $k(x_i, x_j)$ defined as kernel function measuring relation between $x_i$ and $x_j$. We choose RBF kernel $k(x_i, x_j) = \exp(-\rho \|x_i - x_j\|^2)$ in this work. Then for a new point $x_{n+1}$, we consider joint distribution of previously and newly sampled points:

$$\begin{pmatrix} \mathbf{f}(x_{1:n}) \\ f(x_{n+1}) \end{pmatrix} \sim \mathcal{N}\left( \begin{pmatrix} \mathbf{m}(x_{1:n}) \\ m(x_{n+1}) \end{pmatrix}, \begin{pmatrix} \mathbf{K}(x_{1:n}) & \mathbf{k}(x_{n+1}; x_{1:n}) \\ \mathbf{k}(x_{n+1}; x_{1:n})^\mathsf{T} & k(x_{n+1}, x_{n+1}) \end{pmatrix} \right), \quad (2)$$

where $\mathbf{k}(x_{n+1}; x_{1:n}) = (k(x_1, x_{n+1}), k(x_2, x_{n+1}), \cdots, k(x_n, x_{n+1}))$. The posterior $f(x_{n+1}|D)$ satisfies Gaussian distribution (Bishop, 2006) with mean and variance as

$$\mu(x_{n+1}|D) = \mathbf{k}(x_{n+1}; x_{1:n})^\mathsf{T} \mathbf{K}(x_{1:n})^{-1} \mathbf{f}(x_{1:n}), \quad (3)$$

$$\sigma^2(x_{n+1}|D) = k(x_{n+1}, x_{n+1}) - \mathbf{k}(x_{n+1}; x_{1:n})^\mathsf{T} \mathbf{K}(x_{1:n})^{-1} \mathbf{k}(x_{n+1}; x_{1:n}), \quad (4)$$

which is the surrogate model of GP. Then we use acquisition function $a(x)$ to choose which point will be evaluated next. Expected improvement (EI) (Mockus, 1975) and upper confidence bound

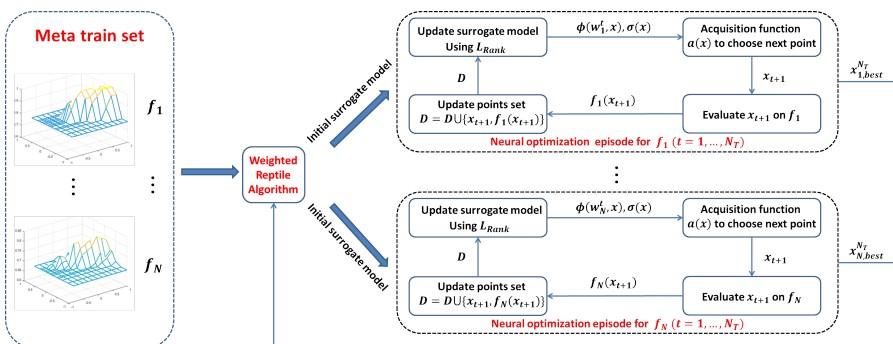

Figure 1: General pipeline of learning initial neural surrogate model using weighted Reptile algorithm from meta train set of functions $f_i, i = 1, \cdots, N$.

(UCB) (Srinivas et al., 2012) have been shown to be efficient. As we aim to find minimizer of $f$, we minimize acquisition function:

$$\arg \min_x a(x) = \arg \min_x \{\mu(x) - \kappa \sigma(x)\} \tag{5}$$

to find next point. As we can see, GP estimates function value via given data $D$ based on surrogate model of Eq. (3) and Eq. (4).

## 2.2 META-LEARNING METHODS

MAML (Finn et al., 2017) and Reptile (Nichol et al., 2018) are two typical meta-learning methods to obtain an initialization of a model from meta train set to well adapt to new problems. Given data set $\mathcal{D}$, model $\mathcal{M}(w)$ and loss function $\mathcal{L}(\cdot, \cdot)$, MAML learns initialization $w^*$ by minimizing $\mathcal{L}(\mathcal{M}(w - \alpha \nabla \mathcal{L}(\mathcal{M}(w), \mathcal{D})), \mathcal{D})$. It implies that MAML learns a $w^*$ that can fast adapt by one gradient descent step. Optimization in MAML uses second-order derivatives of $\mathcal{L}(w)$. In our application, Bayesian optimization procedures are non-differentiable. Therefore, MAML can not be applied to our task directly. Reptile is one of first-order meta-learning algorithms. As illustrated in Alg. 1, to obtain an optimal initialization of model parameters $w$ that can adapt well to new problem $\tau$ by $k$-steps of optimization denoted by $U_\tau^k(w)$, the parameter $w$ is iteratively updated by vector $(\widetilde{w} - w)$ that enforces $w$ to be close to the updated parameters $\widetilde{w}$ of different tasks. In this work, we borrow idea from Reptile, but design a weighted Reptile algorithm for learning initial network-based surrogate model.

---

**Algorithm 1** Reptile

---

1: Initialize $w$, i.e., the initial parameter
2: **for** $i = 1, 2, \cdots, n$ **do**
3:     Sample task $\tau$
4:     Compute $\widetilde{w} = U_\tau^k(w)$, i.e., $k$ steps optimization by SGD or Adam
5:     Update $w \leftarrow w + \epsilon(\widetilde{w} - w)$
6: **end for**

---

## 3 METHOD

We aim to learn to warm-starting Bayesian optimization by substituting mean in Gaussian process by a network-based surrogate model $\phi(w, x)$. Then we learn an initialization of the neural surrogate model from a meta train set of related problems using a *weighted Reptile algorithm*, for adaptation to new problem. We first present some key definitions.

***Neural surrogate model.*** The surrogate model of Bayesian optimization is Eqs. (3) and (4), where mean in Eq. (3) is the expectation of predicted function value and variance in Eq. (4) represents uncertainty. Since related problems in meta train set may have related shapes of function surfaces,

we substitute the mean by a learnable network $\phi(w, x)$ to learn initial surrogate model from meta train set. We still use the variance in Eq. (4) to measure uncertainty, because the estimation uncertainty should be independent in individual problems. Using network for modeling surrogate model, we learn an initial network $\phi(w^*, x)$ from meta train set that can be adapted to new problem. Our neural surrogate model is just the combination of $\phi(w, x)$ and variance in Eq. (4).

***Neural optimization episode.*** An *optimization episode* is the iterative process for optimizing the black-box function. For a problem $Q$ with corresponding black-box function $f$, in $(t + 1)$-th step, we already have surrogate network $\phi(w^t, x)$ and observed points set $\mathcal{D} = \{x_j, f(x_j)\}_{j=1}^t$. First, we update neural surrogate model using $\mathcal{L}_{Rank}$ (will be discussed in sect. 3.1) and Eq. (4), and obtain $\phi(w^{t+1}, x)$ and $\sigma^2(x)$. Then we use acquisition function of Eq. (5) to choose next point $x_{t+1}$. After that, we update points set $\mathcal{D} = \mathcal{D} \bigcup \{x_{t+1}, f(x_{t+1})\}$. We iterate the above process to obtain $x_{best}^{N_T} = \arg\min_{\{x_n, n=1,...,N_T\}} \{f(x_n)\}$. We called this procedure as *Neural optimization episode (NOE)*, as illustrated in the right part of Fig. 1.

In this work, we aim to learn initial $\phi(w, x)$ to initialize NOE of a new problem given a set of related problems in meta train set, as illustrated in Fig. 1. Since the dynamics of NOE is complicated and might be non-differentiable, we propose a weighted Reptile Algorithm to learn the initial $\phi(w, x)$ from meta train set using reward defined based on $x_{best}^{N_T}$ of problems. This learned initial model $\phi(w, x)$ is utilized to initialize the neural surrogate model of a new problem for its NOE.

### 3.1 RANKING LOSS FOR LEARNING NEURAL SURROGATE MODEL

We now focus on how to update neural surrogate model given observed points and their function values in neural optimization episode. As we know, different problems have different ranges of function values. To make surrogate model insensitive to the specific range of function values, our neural surrogate model is learned to predict ranking of function values of points instead of exactly fitting the function values. Describing data sets based on pairwise rankings has been investigated in a few machine learning approaches (Leite et al., 2012; Wistuba et al., 2015; Zheng et al., 2007).

Given points set $\mathcal{D} = \{x_i, f(x_i)\}_{i=1}^n$, we update $w$ of neural surrogate model $\phi(w, \cdot)$ by $\mathcal{L}_{Rank}$. First we sort points in $\mathcal{D}$ by $\{f(x_i)\}_{i=1}^n$ in descent order and get $\mathcal{D}_s = \{x_{r(i)}, f(x_{r(i)})\}_{i=1}^n$, where $r(i), i = 1 \cdots n$ is a permutation of $1 \cdots n$. Then we utilize $\mathcal{L}_{Rank}$ as loss:

$$\mathcal{L}_{Rank}(\phi(w), \mathcal{D}) = \sum_{i=1}^n \sum_{j=1, j>i}^n [\max(\phi(w, x_{r(j)}) - \phi(w, x_{r(i)}) + \delta, 0)]^2, \tag{6}$$

$\mathcal{L}_{Rank}$ is based on hinge loss, and minimization of $\mathcal{L}_{Rank}$ w.r.t. $w$ enforces that $\phi(w, x_{r(i)})$ is strictly larger than $\phi(w, x_{r(j)})$ with margin of $\delta$. Note that final layer of our surrogate model $\phi$ is a sigmoid function, therefore, the output of surrogate model is constrained in range of $(0, 1)$ irrespective of different problems. We use gradient descent method to optimize $\mathcal{L}_{Rank}$, please see appendix A.1 for detail. We denote $GD(w, \mathcal{L}_{Rank})$ as the learned surrogate model parameters by minimizing $\mathcal{L}_{Rank}$ using gradient descent method. This above process is taken as the updating step of neural surrogate model in neural optimization episode.

### 3.2 WEIGHTED REPTILE ALGORITHM FOR LEARNING INITIAL SURROGATE MODEL

We next focus on how to learn an initial neural surrogate model from a set of related problems in meta train set, taken as the initialization of neural surrogate model for NOE of a new problem. We propose a weighted Reptile algorithm to train the initial neural surrogate model.

Reptile in Alg. 1 is an effective meta-learning algorithm that can learn a good initialization of model parameter from a meta train set. Inspired by Reptile, our *Weighted Reptile Algorithm* samples a set of initializations of surrogate model parameters for NOE of each problem in the meta train set for exploration, and these sampled initializations are selected by reward and utilized for parameter updating with weights defined by rewards.

**Reward.** Given minimizer $x_i^*$ of $i$-th problem $Q_i(x)$ in meta train set, reward at step $t$ of NOE of problem $Q_i(x)$ is defined as $r_i^t \doteq -\|x_{i,best}^t - x_i^*\|^2$, where $x_{i,best}^t = \arg\min_{\{x_n, n=1,...,t\}} \{f_i(x_n)\}$,

---

**Algorithm 2** Neural optimization episode (NOE).

---

**Require:** Maximum iterations $N_T$, $\kappa > 0$, test problem $\mathcal{Q}$ with function $f$, initialized parameter $w$
**Ensure:** $x_{best}, w^{N_T}$
  1: set $w^1 = w, \mathcal{D} = \{\}$
  2: $x^1 = \arg\min_x \phi(w^1, x)$
  3: $y^1 = f(x^1), \mathcal{D} = \mathcal{D} \bigcup \{x^1, y^1\}$
  4: Update $\sigma(x)$ by Eq. (4)
  5: **for** $t = 1 : N_T - 1$ **do**
  6:    $x^{t+1} = \arg\min \phi(w^t, x) - \kappa\sigma(x)$
  7:    $y^{t+1} = f(x^{t+1}), \mathcal{D} = \mathcal{D} \bigcup \{x^{t+1}, y^{t+1}\}$
  8:    $w^{t+1} = GD(w^t, \mathcal{L}_{Rank})$
  9:    Update $\sigma(x)$ by Eq. (4)
10: **end for**
11: $x_{best} = x_{t^*}, t^* = \arg\min_t \{y^t\}_{i=t}^{N_T}$
12: **return** $x_{best}, w^{N_T}$

---

$f_i$ is the black-box function in problem $Q_i(x)$, $x_n$ is sampled point at $n$-th step, $x_{i,best}^t$ is the best point up to step $t$.

**Weighted Reptile Algorithm.** In Reptile, as in line 5 of Alg. 1, the parameter $w$ is learned to be close to updated parameter $\widetilde{w}$ by adapting $w$ in different problems. It implicitly assumes that updated parameter $\widetilde{w}$ should be better than initialized $w$ for corresponding problem. In our approach, we aim to learn initial surrogate model $\phi(w, x)$ from problems in meta train set. Similar to Reptile, we learn initial network parameter $w$ of $\phi(w, x)$ close to updated parameters after running NOE of problems in meta train set. To explore diverse paths of NOE, for each $i$-th problem in meta train set, we sample a few ($M$) initial parameters $\{w_i^p\}_{p=1}^M \sim \mathcal{N}(w, \sigma^2)$ by disturbing currently estimated $w$. Then after running $N_T$ steps of NOE, we obtain points $\{x_{i,best}^{N_T,p}\}_{p=1}^M$ and updated parameters $\{\widetilde{w}_i^{N_T,p}\}_{p=1}^M$ starting from surrogate models initialized with these sampled parameters. To learn a good initial parameter $w$ of surrogate model $\phi$, parameter is updated by $w = w + \beta\Delta w$, where

$$\Delta w \;\; = \;\; \sum_{p=1}^M \sum_{i=1}^N (\widetilde{w}_i^{N_T,p} - w) \max(r_{i,p}^{N_T} - r_i^{N_T}, 0), \tag{7}$$

and reward $r_i^{N_T} = -\|x_{i,best}^{N_T} - x_i^*\|^2, r_{i,p}^{N_T} = -\|x_{i,best}^{N_T,p} - x_i^*\|^2$. For $i$-th problem and $p$-th sampled parameter, weight $\max(r_{i,p}^{N_T} - r_i^{N_T}, 0)$ is always non-negative, and it is non-zero only when the sampled parameter $w_i^p$ can produce better reward than the parameter $w$ itself if taking them as initial surrogate model parameters for running NOE of problem $Q_i$. This rule enforces that the parameter $w$ of our learned initial surrogate model is close to better adapted parameters of different problems in meta train set. Algorithm 3 presents the pipeline of meta-train process using weighted Reptile algorithm, and it depends on the NOE as described in Alg. 2 .

## 4 EXPERIMENTS

We evaluate our method for global optimization on synthetic functions set, and hyper-parameters tuning in evolutionary algorithm, neural network and SVM. In the following paragraphs, our method based on neural surrogate model and weighted Reptile algorithm will be dubbed as *WRA-N*. In all experiments, we set $\kappa = 0.1$, $\delta = 0.01$, $\sigma = 0.1$, training epoch $N_E = 30$, learning rate $\beta = 0.1$, number of samples $M = 50$. We use the mean squared error (MSE) between function values of our found best point and the global minimizer to measure the performance of optimization. We mainly compare with TST-R (Wistuba et al., 2016) and GP-UCB (Srinivas et al., 2012). Moreover, we also compare with meta feature-based method TST-M (Wistuba et al., 2016) in SVM hyper-parameter tuning. But for the remaining tasks, we did not compare with it because it is hard to well define meta-features for these problems. The neural surrogate model is taken as a network with one hidden layer of 15 hidden units, and both hidden and output layers are with sigmoid nonlinearity.

---

**Algorithm 3** Meta train by weighted Reptile algorithm.

---

**Require:** Number of iterations $N_T$ in NOE, $\kappa > 0$, meta train set $\mathcal{Q}$, number of epochs $N_E$, learning rate $\beta$, number of samples $M$, standard deviation $\sigma$ for sampling.
**Ensure:** $w^*$
1: Randomly initialize $w^1$
2: *#Meta-train loop.*
3: **for** $l = 1 : N_E$ **do**
4:    *#Surrogate model adaptation loop.*
5:    **for** $i = 1 : |\mathcal{Q}|$ **do**
6:       $\{x_{i,best}^{N_T}, w_i^{N_T}\} = \text{NOE}(N_T, \kappa, \mathcal{Q}_i, w^l)$
7:       $r_i^{N_T} = -\|x_{i,best}^{N_T} - x_i^*\|^2$
8:       **for** $p = 1 : M$ **do**
9:          set $\widehat{w}_i^p = w^l + \mathcal{N}(0, \sigma^2 I)$
10:          $\{x_{i,best}^{N_T,p}, \tilde{w}_i^{N_T,p}\} = \text{NOE}(N_T, \kappa, \mathcal{Q}_i, \widehat{w}_i^p)$
11:          $r_{i,p}^{N_T} = -\|x_{i,best}^{N_T,p} - x_i^*\|^2, g_i^p = \tilde{w}_i^{N_T,p} - w^l$
12:       **end for**
13:    **end for**
14:    *#Updating initial surrogate model parameter.*
15:    $w^{l+1} = w^l + \beta \sum_{i=1}^{|\mathcal{Q}|} \sum_{p=1}^{M} \max(r_{i,p}^{N_T} - r_i^{N_T}, 0) g_i^p$
16: **end for**
17: **return** $w^* = w^{N_E+1}$

---

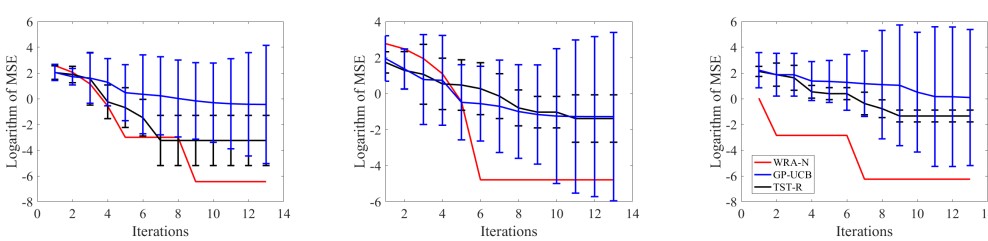

Figure 2: MSEs w.r.t. optimization iterations on synthetic functions with $a = \frac{3\pi}{2}, \pi, \frac{2\pi}{3}$.

## 4.1 SYNTHETIC FUNCTIONS

We first evaluate on Alpine 1 function in Jamil & Yang (2013) without using absolute value: $f(x) = x sin(x + a) + 0.1x, x \in [-10, 10]$. We set functions with $a = \frac{4\pi}{3}, \frac{5\pi}{4}, \frac{6\pi}{5}, \frac{7\pi}{6}$ as meta train set, and 18 functions with $a = \frac{3\pi}{2} - \frac{i\pi}{24}$, $i \in \{1, ..., 21\}$ excluding $\{4, 6, 8\}$ in meta train set for testing. Figure 2 shows logarithm of MSEs w.r.t. optimization iterations of WRA-N, TST-R and GP-UCB for $a = \frac{3\pi}{2}, \pi, \frac{2\pi}{3}$, and other results are in appendix. Optimization iteration means the iteration in optimization episode, e.g., the iterations in NOE of our WRA-N. Since TST-R needs base models for combination, we sample 20 points from uniform distribution in $(-10, 10)$ to construct base models. TST-R and GP-UCB start with 3 random points, then run in 10 iterations. Our WRA-N starts with neural surrogate model learned by meta train set, and runs in 13 iterations, i.e., $N_T = 13$. Optimization was repeated 20 times for TST-R and GP-UCB. For TST-R, each time is with differently sampled points for constructing base models. Figure 2 shows curves of average MSEs with variances in logarithm w.r.t. iterations. WRA-N starts with learned initial surrogate model, therefore curves are deterministic. WRA-N significantly improves baseline GP-UCB and outperforms TST-R, because test functions may not be well represented by base functions in TST-R.

## 4.2 HYPER-PARAMETERS TUNING IN DIFFERENTIAL EVOLUTIONARY ALGORITHM

Differential evolution (DE) (Galletly, 1996; Das & Suganthan, 2011) is a global optimization algorithm to minimize function $f : R^d \to R^1$. DE starts with initial population $X^0 =$

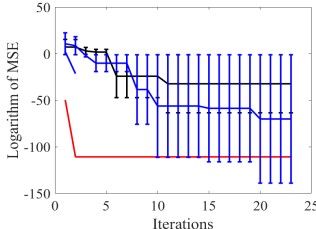 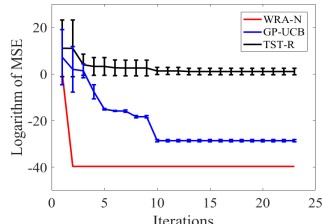

Figure 3: MSEs w.r.t. iterations for $f_5$, $f_6$ in CEC 2017 for hyper-parameters tuning of DE.

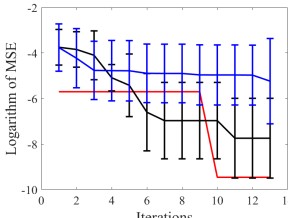 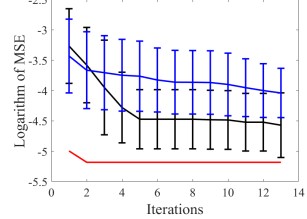 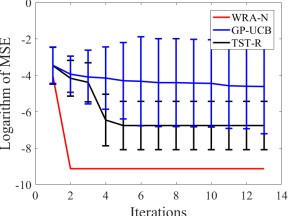

Figure 4: MSEs w.r.t. optimization iterations for hyper-parameters tuning of neural networks on data sets of $\mathcal{S}_4, \mathcal{S}_5, \mathcal{S}_6$.

$[x_1^0, x_2^0, \cdots, x_{N_p}^0] \in R^{d \times N_p}$, and iterates $u_i^{k+1} = x_i^k + F(x_{best}^k - x_i^k) + F(x_{r(1)}^k - x_{r(2)}^k)$, where $x_{r(i)}^k (i = 1, 2)$ are randomly chosen from $X^k$, $x_{best}^k$ is the best point in $k$-th iteration, $F$ is hyper-parameter. Then we can use binomial crossover and selection methods to create $X^{k+1}$. We generalize this iteration formulation to be

$$u_i^{k+1} = x_i^k + F(x_{best}^k - x_i^k) + D(x_{r(1)}^k - x_{r(2)}^k)$$

using 2 hyper-parameters $F$ and $D$, which is a general formulation covering most DE formulations. $F$ and $D$ influence optimization performance of DE (e.g., finding minimum of functions by DE) significantly, and thus we can utilize our method to tune these hyper-parameters. We use function set of CEC 2017 (Awad et al., 2016) for optimization by DE and use our method to find optimal hyper-parameters. The neural surrogate model is to predict the function values w.r.t. hyper-parameters. We set function dimension as 10, construct meta train set containing basic functions of $f_1, f_2, f_3, f_4$, and learn initial surrogate model that can find optimal hyper-parameters well in $f_5 - f_{10}$ for testing. The formulas of these functions are shown in appendix. For TST-R, we sample 20 points uniformly in feasible domain and use these points to construct base models. TST-R and GP-UCB start with 3 randomly chosen points, and then proceed sequentially for 20 iterations. Our method starts with surrogate model learned by meta train set, and proceeds for a total of 23 iterations. The optimization was repeated 20 times similar to the experiments on synthetic function set. Figure 3 shows results on $f_5$ and $f_6$, and results on the other functions are shown in appendix. Our method produces best hyper-parameter tuning performance. Note that the functions in meta train set are in different scales of function values. For TST-R, we follow its original implementation by normalizing point values of base models to normal distribution. However, in our approach, the neural surrogate model is updated using ranking-based loss, therefore it is not sensitive to the different scales of functions.

## 4.3 HYPER-PARAMETERS TUNING IN NEURAL NETWORK

We will optimize hyper-parameters of neural networks for image classification. We use EMNIST data set (Cohen et al., 2017), i.e., an extension of MNIST to handwritten letters. There are 62 classes of [0-9], [a-z] and [A-Z], and several classes that are similar between uppercase and lowercase are merged, and finally we have 47 classes in data set. The network is with one hidden layer, sigmoid as activation function and softmax for classification, followed by cross-entropy loss. We first randomly sample 10 classes in 3 times from EMNIST with 1000 images for each class,

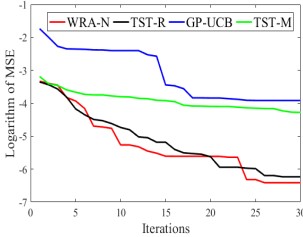 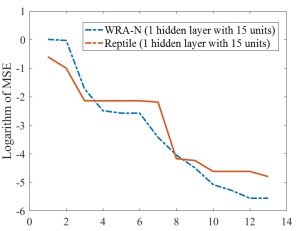 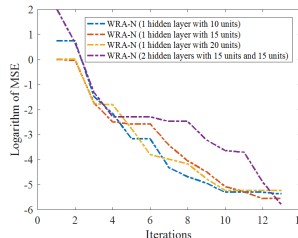

Figure 5: MSEs w.r.t. iterations for hyper-parameters tuning of SVMs.

Figure 6: Ablation study.

and construct sets of $\mathcal{S}_i, i = 1, 2, 3$ as meta train sets. From the remaining classes, we construct test sets by randomly sampling 10 classes with 1000 images for each class in 8 times and produce $\mathcal{S}_4 - \mathcal{S}_{11}$. We aim to tune the number of hidden units $N_h$ and learning rate $lr$ of S-GD as hyper-parameters. The candidate sets of $N_h$ and $lr$ are respectively $\{10, 15, \cdots, 200\}$ and $\{0.001, 0.002, \cdots, 0.01, 0.02, 0.03, \cdots, 0.1, 0.2, 0.3, \cdots, 1, 2, 3, \cdots, 10\}$. For this task, training accuracy is higher with increase of number of hidden units because more units produce higher capacity but may over-fit training data. Thus we define a criterion to balance accuracy (denoted by AC) and model capacity determined by $N_h$ using ERR $= (N_h + c)/\text{AC}$ where $c$ is the trade-off coefficient and set to 1000. Then neural surrogate model is to predict ERR w.r.t. to hyper-parameters. Note that we can also use other criterion for learning hyper-parameters. Figure 4 shows MSEs (between ERRs of best hyper-parameters and estimated hyper-parameters) w.r.t. optimization iterations by different methods. We can observe that our method performs best for this task.

### 4.4 HYPER-PARAMETERS TUNING IN SVM

Support vector machine (SVM) dataset (Wistuba et al., 2016) consists of 50 classification datasets randomly chosen from UCI repository. The SVMs use Gaussian kernel with hyper-parameters of trade-off parameter $C$ and width $\gamma$ of Gaussian kernel. The candidate hyper-parameters are $C \in \{2^{-5}, ..., 2^{6}\}$ and $\gamma \in \{10^{-4}, 10^{-3}, 10^{-2}, 0.05, 0.1, 0.5, 1, 2, 5, 10, 20, 50, 10^{2}, 10^{3}\}$. The neural surrogate model is to predict misclassification rate w.r.t. to hyper-parameters $C, \gamma$. In experiments, we randomly choose 5 datasets as meta train sets, and the others as test sets. All the experiments are performed in 10 times. Figure 5 shows the average MSEs w.r.t. optimization iterations by different methods. Here we compare our WRA-N with TST-M (Wistuba et al., 2016), which also relies on linear combination of surrogate models of meta train problems but using meta features. We can see that our WRA-N performs better than both TST-R and TST-M. Both our WRA-N and TST-R perform significantly better than baseline GP-UCB.

### 4.5 ABLATION STUDY

We now justify effectiveness of weighted Reptile algorithm (WRA) and different network structures of surrogate model $\phi$. All following networks are evaluated on synthetic set in section 4.1. In Fig. 6 (left), we compare our WRA with traditional Reptile algorithm without using sampling and weighting strategy. The average MSE curves show that our WRA performs better than Reptile. We further compare different structures of surrogate models including networks with one hidden layer of 10, 15, 25 units and 2 hidden layers with 15 units. We observe that they perform similarly for networks with one hidden layer, and adding more layers does not apparently improve performance.

## 5 CONCLUSION

We have proposed a novel neural surrogate model and a weighted Reptile algorithm to learn an initial neural surrogate model for warm-starting Bayesian optimization. Extensive experiments on synthetic functions optimization and hyper-parameters tuning in multiple tasks have shown promising performance. In the future, we are interested to apply it to more global optimization or hyper-parameters tuning tasks, and analyze its theoretical properties.

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

# A    APPENDIX

In the appendix, we will first introduce how to optimize the ranking loss in A.1. Second, we will show the functions in CEC 2017 for DE hyper-parameters tuning in A.2. Third, in A.3, we will show more experimental results for synthetic function optimization, hyper-parameters tuning in DE and neural network. Fourth, we will show example to illustrate the neural surrogate model optimized by minimizing $\mathcal{L}_{Rank}$ in A.4. Fifth, we will show the training curves of our weighted Reptile algorithm for experiments in A.5. Finally, we will discuss on the relation between neural surrogate model and Gaussian process in A.6.

## A.1    GRADIENT METHOD FOR MINIMIZING $\mathcal{L}_{Rank}$

We first recall the definition of $\mathcal{L}_{Rank}$ in section 3.1. Given points set $\mathcal{D} = \{x_i, f(x_i)\}_{i=1}^n$, we sort points in $\mathcal{D}$ by $\{f(x_i)\}_{i=1}^n$ in descent order and get $\mathcal{D}_s = \{x_{r(i)}, f(x_{r(i)})\}_{i=1}^n$, where $r(i), i = 1 \cdots n$ is a permutation of $1 \cdots n$. Then the $\mathcal{L}_{Rank}$ is defined as

$$\mathcal{L}_{Rank}(\phi(w), \mathcal{D}) = \sum_{i=1}^n \sum_{j=1, j>i}^n ([\phi(w, x_{r(j)}) - \phi(w, x_{r(i)}) + \delta]_+)^2,$$

where $[q]_+ = \max(q, 0)$. Then $GD(w, \mathcal{L}_{Rank})$ is a gradient based method that iteratively runs the following procedure. Given each pair of sampled points $(x_{r(i)}, x_{r(j)})$, we update parameter $w$ of surrogate model by

$$w^{t+1} = w^t - \eta[\phi(w^t, x_{r(j)}) - \phi(w_t^n, x_{r(i)}) + \delta]_+(\nabla\phi(w_t^n, x_{r(j)}) - \nabla\phi(w_t^n, x_{r(i)})),$$

where $\eta$ is step size.

## A.2 FUNCTIONS IN CEC 2017 FOR DE HYPER-PARAMETERS TUNING

The ten functions are listed as follows.

$$
f_1(x) \quad = x_1^2 + 10^6 \sum_{i=2}^{H} {x_i}^2, f_2(x) = \sum_{i=1}^{H} |x_i|^{i+1}
$$

$$
f_3(x) \quad = \sum_{i=1}^{H} {x_i}^2 + (0.5 \sum_{i=1}^{H} x_i)^4 + (0.5 \sum_{i=1}^{H} x_i)^4
$$

$$
f_4(x) \quad = \sum_{i=1}^{H-1} (100({x_i}^2 - x_{i+1})^2 + (x_i - 1)^2)
$$

$$
f_5(x) \quad = \sum_{i=1}^{H} ({x_i}^2 - 10cos(2\pi x_i) + 10)
$$

$$
f_6(x) \quad = g(x_1, x_2) + g(x_2, x_3) + \cdots + g(x_H, x_1)
$$

$$
f_7(x) \quad = \min(\sum_{i=1}^{H} (\hat{x}_i - \mu_0)^2, dH + s \sum_{i=1}^{H} (\hat{x}_i - \mu_1)^2) + 10(H - \sum_{i=1}^{H} cos(2\pi \hat{z}_i))
$$

$$
f_8(x) \quad = \sum_{i=1}^{H} ({z_i}^2 - 10cos(2\pi z_i) + 10)
$$

$$
f_9(x) \quad = sin^2(\pi w_1) + \sum_{i=1}^{H-1} (w_i - 1)^2[1 + 10sin^2(\pi w_i + 1)] + (w_H - 1)^2[1 + sin^2(2\pi w_H)]
$$

$$
f_{10}(x) \quad = 418.9829H - \sum_{i=1}^{H} g(z_i)
$$

$g(x, y)$ in $f_6$ is defined as $g(x, y) = 0.5 + \frac{sin^2(\sqrt{x^2+y^2})-0.5}{(1+0.001(x^2+y^2))^2}$. Please refer to CEC 2017 (Awad et al., 2016) for details on parameters.

## A.3 ADDITIONAL RESULTS IN EXPERIMENTS

Figure 7 shows the function optimization results on 15 remaining test functions in synthetic function set. Figure 8 shows curves of average MSEs w.r.t. different optimization iterations for tuning hyper-parameters of number of hidden units and learning rate of SGD on EMNIST data sets of $\mathcal{S}_7, \mathcal{S}_8, \mathcal{S}_9, \mathcal{S}_{10}, \mathcal{S}_{11}$. Figure 9 shows these curves for tuning hyper-parameters in DE on functions of $f_7 - f_{10}$ in CEC 2017.

## A.4 AN ILLUSTRATION OF NEURAL SURROGATE MODEL BY MINIMIZING $\mathcal{L}_{Rank}$

Figure 10 shows the neural surrogate model by minimizing $\mathcal{L}_{Rank}$ working on DE hyper-parameters tuning. In this task, we use neural surrogate model to predict the mapping from hyper-parameters $D, F$ to function values of points. Figure 10(a) shows the surrogate model learned by neural network. Figure 10(b) shows the real function values w.r.t. hyper parameters. Figure 10(c) shows the logarithm of real function values w.r.t. hyper-parameters. Then we see that the neural surrogate model is within range of [0, 1] and it has similar local minimum with real function (by comparing Figs. 10(a) and (c)), but it is more smooth than real function surface. We further illustrate the distributions using histograms of function values of real function and neural surrogate model. As shown in Fig. 10 (d), x-axis is the percentage (i.e. ratio) of maximal function value, y-axis is the percentage of number of data with function values falling in these ratios to maximal function values. We see that, for real function, most values are less than 0.25 of maximum function value, while our neural surrogate model is more uniform.

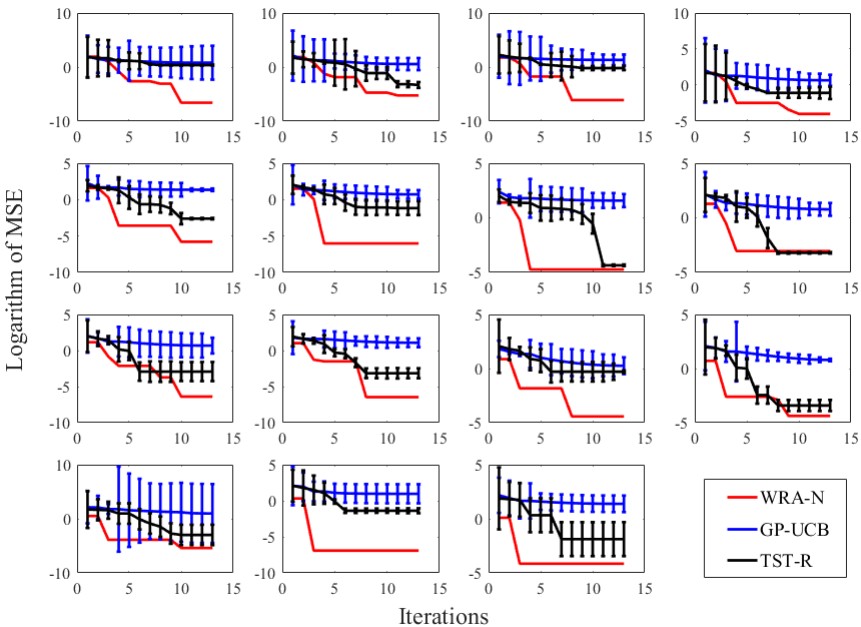

Figure 7: MSEs (in logarithm) w.r.t. different optimization iterations on 15 remaining test functions in synthetic function set.

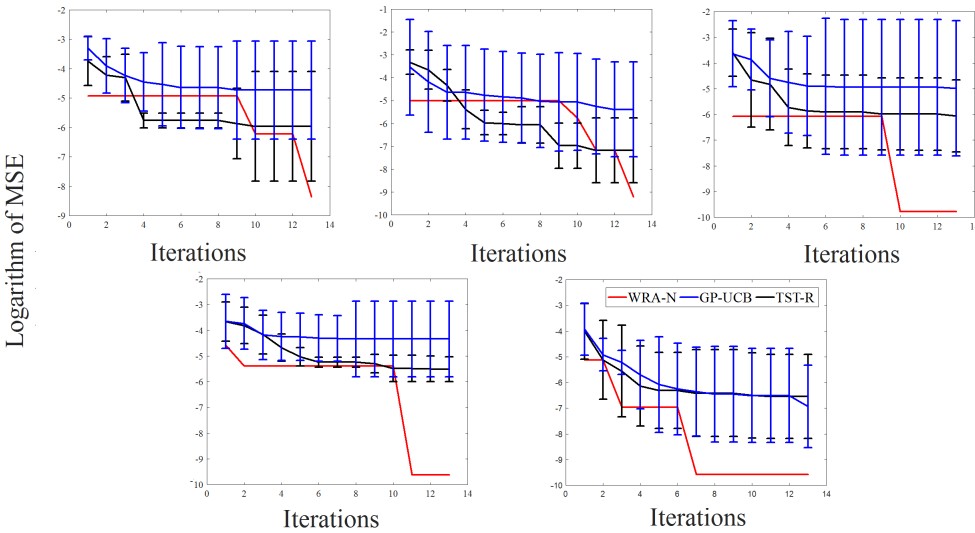

Figure 8: MSEs (in logarithm) w.r.t. different optimization iterations for tuning number of hidden units and learning rate on EMNIST data sets of $\mathcal{S}_7, \mathcal{S}_8, \mathcal{S}_9, \mathcal{S}_{10}, \mathcal{S}_{11}$.

## A.5 THE TRAINING CURVES OF OUR WEIGHTED REPTILE ALGORITHM

In the four experiments in section 4, we use our weighted Reptile algorithm to learn the initial neural surrogate model. We now show the training curves when training initial surrogate model from meta train set. In Fig. 11, we show the curves of total rewards of $r_i^{N_T}$ on all problems in meta train set for four experiments. They show that weighted Reptile algorithm does increase the reward when training the initial surrogate model with increase of training epochs.

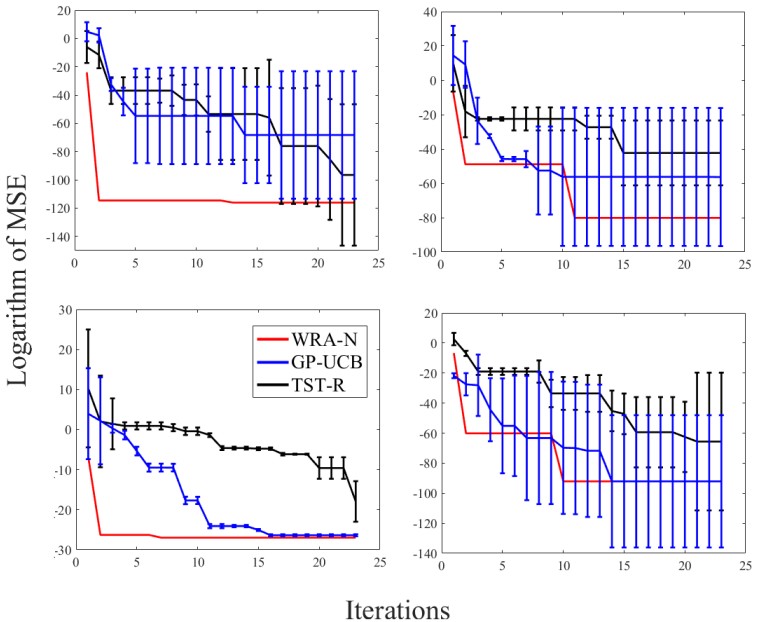

Figure 9: MSEs (in logarithm) w.r.t. different optimization iterations for tuning hyper-parameters of DE on $f_7 - f_{10}$ of CEC 2017.

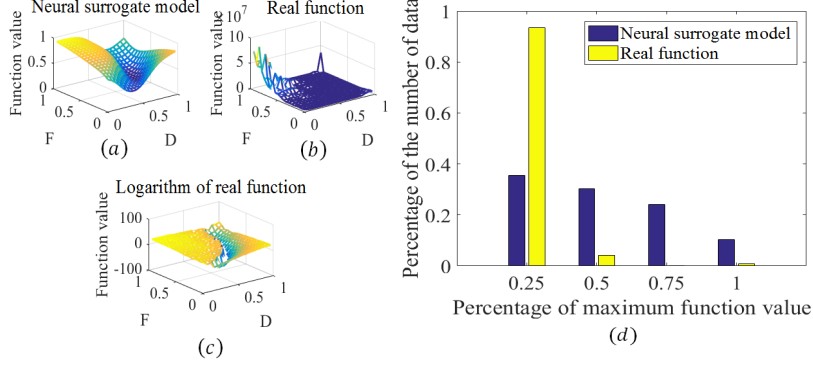

Figure 10: An Illustration of neural surrogate model for hyper-parameter tuning of DE.

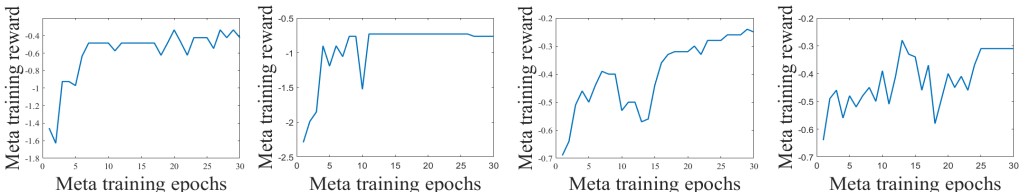

Figure 11: Training curves of rewards on meta train set respectively for our four different experiments in sections 4.1-4.4.

## A.6 RELATION BETWEEN NEURAL SURROGATE MODEL AND GAUSSIAN PROCESS

Neal (2012) has proved that a network with random weights and one hidden layer converges to a Gaussian process when the number of hidden units increases. In this work, we directly model the surrogate model using neural network by substituting the formulation of function value prediction in Gaussian process. Different to Gaussian process, our network is with finite number of hidden units and learnable parameters, facilitating effective model learning for the task of learning to initialize surrogate model from a meta train set.

