# OpenReview forum: "Learning Neural Surrogate Model for Warm-Starting Bayesian Optimization"
_ICLR.cc/2020/Conference — Reject_

### Official Review · AnonReviewer3 · 2019-10-20
**Official Blind Review #3**

**Rating:** 3

**Review:**

Learning from past experience to quickly adapt to a new task has been an important and fast-growing issue in machine learning. Such technique facilitates Bayesian optimization as well, warm-starting Bayesian optimization. Recently a few methods have been developed along this direction, from designing handcrafted meta-features to learning meta-features. The current paper takes a similar step, learning neural surrogate model from related tasks to warm-start Bayesian optimization. The main idea is to replace the mean function of GP by neural surrogate model, so that parameterized models are used for meta-training, in the framework of RETILE. An idea of weighted REPTILE is another contribution in this paper, where parameter updates are done with weighs defined by rewards.

---Strength---

- Learning initialization for a surrogate model to warm-start Bayesian optimization is a sound approach.

---Weakness---

- While mean function of GP is replaced by a neural surrogate model, the posterior variance of GP should be calculated. In other words, GP regression should be run in addition to updating the neural surrogate model. One can use the conditional neural process (instead of GP regression). Have a look at the ICML18 paper: Marta Garnelo et al. (2018), "Conditional neural processes," ICML.

---Comments---

- You can also learn an initial mean function of GP. Any comparison?
- There is also interesting work on meta Bayesian optimization: Zi Wang et al. (2018), "Regret bounds for meta Bayesian optimization," NeurIPS.
- Ranking loss is used to train neural surrogate models. It is not clear why minimizing ranking loss makes sense in this case. It will be different from the mean function of GP regression, so it is not clear what is the behavior of the acquisition function constructed by the neural surrogate model as the posterior variance of GP.


**Experience Assessment:**

I have published in this field for several years.

**Review Assessment: Checking Correctness Of Derivations And Theory:**

I assessed the sensibility of the derivations and theory.

**Review Assessment: Checking Correctness Of Experiments:**

I assessed the sensibility of the experiments.

**Review Assessment: Thoroughness In Paper Reading:**

I read the paper at least twice and used my best judgement in assessing the paper.

---

> ### Author Response · Authors · 2019-11-15
> **Rebuttal**
>
> Thanks for your review,
> I have read the paper you mentioned in ICML 2018  Marta Garnelo et al. (2018). It seems that this paper learned a conditional process. However, it seems learning a process based on several functions in training set, which means that it needs much sampling points to train. In real problems, we can only sample few points on related problems. Thus this work cannot be used here.
>
> Ranking loss aims to eliminate the scale problem in real problems. Though we can standardize data in train set, we cannot standandize data in test set.

---

### Official Review · AnonReviewer2 · 2019-10-21
**Official Blind Review #2**

**Rating:** 1

**Review:**

Summary of the paper:

The authors propose a meta-learning approach for BO. The method consists in using a NN as the predictive mean of the model used to guide the search in BO. This NN is initialized cleverly so that its solutions is close to the actual solution to the problem by using related optimization tasks. The proposed method is validated on several experiments.

Detailed comments:

The writing of the paper needs to be improved. It has awkward sentences like "Bayesian optimization iteratively samples new point by".

The intro on Bayesian optimization has to be improved, it explain very poorly this technique.

The description of the MAML method is not clear. The same for the reptile algorithm.

Eq. (3) is wrong. It does not take into account that the mean of the GP is different from 0.

It seems the authors replace the mean of the GP predictive distribution with the output of a neural network. This is strange and not very well justified. I would have expected that they use the output of the NN as the GP prior mean, to then compute the GP posterior mean.

In the related problems the actual objective is unknown. How is that difficulty addressed?

In Section 3.1, why do not you standardize the output values to have zero mean and unit deviation instead of using the ranking?

How are the hyper-parameters of the GP tuned? It seems GP based method will fail, essentially, because the authors do not consider the posterior distribution of the GP and change the mean of the predictive distribution to be the NN output.

There are no error bars for WRA-N in the experiments. Does this means that only one realization has been carried out? If so, this is insufficient to extract any conclusion. The results can be simply obtained by chance. The authors should consider several repetitions with different random seeds or different problems.

The experiments in section 4.1 are non realistic, since the actual shape of the objective function is known beforehand.

In section 4.2 it is not clear what relation are between the train / test functions. Therefore, it is not possible to understand why the proposed approach works better.

The experiments need another baseline to compare with. Namely, the same method in which the NN is randomly initialized. This will allow to check that the meta-learning procedure is useful. Currently, it can be the case that the improvements are simply due to using a different model for optimization.

Summing up, I think that:

(1) This paper needs further improvement in the writing.

(2) The experimental section is questionable since there are missing methods in the comparison and no error bars in the experiments.

Therefore I believe that this paper is still at an early stage and not ready for publication.


**Experience Assessment:**

I have published one or two papers in this area.

**Review Assessment: Checking Correctness Of Derivations And Theory:**

I did not assess the derivations or theory.

**Review Assessment: Checking Correctness Of Experiments:**

I assessed the sensibility of the experiments.

**Review Assessment: Thoroughness In Paper Reading:**

I read the paper at least twice and used my best judgement in assessing the paper.

---

> ### Author Response · Authors · 2019-11-15
> **Rebuttal**
>
> Thanks for your review,
> 1 ‘In the related problems the actual objective is unknown. How is that difficulty addressed?',  We don't need real objective,  just a few number of sampling points.
> 2 You can standardize data in  train set but how to standardize data in test set? We only observe few points, standardize is not valid.
> 3 ‘How are the hyper-parameters of the GP tuned? It seems GP based method will fail, essentially, because the authors do not consider the posterior distribution of the GP and change the mean of the predictive distribution to be the NN output’ I cannot understand what you are talking about.
> 4 The randomness of bayesian optimization comes from the randomly sampled initial points. WRA-N doesn't have error bar as it is determinate as we sampled first point by minimizing neural surrogate model.
> 5 'experiments in section 4.1 are non realistic' , we propose a warm starting bayesian 'optimization' method. I don't think it is questionable to optimize a synthetic function. Futhermore, the rest of 3 experiments are real problems.
> 6 CEC2017 consists of several test functions, which are used to test efficiency of evolutionary algorithms, just like image-net. And DE is a traditional method using on this dataset. When using an algorithm, hyper-parameters cannot be sensitive (there is a small gap of hyper-parameters between different functions). Thus we set a part of funcitons in CEC2017 to train and others to test.
> 7 I can give you the comparation about initialization on ablation study (on problem of synthetic function):
> initialization    |2.1|1.2|  0  |-0.8|-1.1|-1.2|-1.3|-1.4|-1.4|-1.4
> after training  |0.1|  0 |-1.9|-2.7|-2.8|-2.8|-3.5|-4.1|-4.4|-5.0

---

### Official Review · AnonReviewer1 · 2019-10-23
**Official Blind Review #1**

**Rating:** 3

**Review:**

POST-REBUTTAL FEEDBACK

Thanks for your response.

The justifications provided in the response have not convinced me to improve my score. They are at times hard to understand: For example, the authors have claimed that while their design choice is not reasonable, it is less unreasonable than the other.

SUMMARY OF REVIEW

The authors have proposed the use of a neural surrogate model in place of the GP posterior mean and a weighted Reptile algorithm to meta-learn the initial weights of the neural surrogate model. This approach appears interesting. However, there seems to be multiple highly restrictive (at times impractical) assumptions in this work that are atypical of the BO setting adopted by other meta BO algorithms and not discussed, as detailed below. Justifications are required.

Clarifications are also needed with regards to how they exactly run their algorithm in the experiments and whether the prior/initial information from related problems/meta tasks provided to the tested algorithms is fair.



DETAILED COMMENTS

The authors say that "We still use the variance in Eq. (4) to measure uncertainty, because the estimation uncertainty should be independent in individual problems." This does not seem to hold true. If a meta task or train set is indeed correlated (or provides information) to the new problem, the posterior variance/uncertainty at a point depends on the observations in the meta task or train set near to this point (see, for example,
Feurer et al. (2018)). Can the authors discuss the implications of such an assumption in their work?

Q and Q_i have always been referred to as problems. In Algorithm 3, Q is suddenly referred to as meta train set. On page 4, you have said that x^*_i is the minimizer of i-th problem Q_i(x) in meta train set. Based on these information, I assume that the authors consider x^*_i as the global minimizer and that x^*_i is known in order to compute the rewards. Can the authors discuss why is this a reasonable assumption?

In their proposed weight Reptile algorithm (Algorithm 3), the authors have also assumed access to the black-box functions of the related problems or meta tasks, which is not typical of other meta BO works that only require the existing observations or datasets of the related problems/meta tasks. As a result, compared with the existing meta BO algorithms, their proposed weighted Reptile is considerably more expensive due to the need to additionally evaluate the black-box functions of the related problems or meta tasks many times during execution. Can the authors discuss the practical implications of such an assumption and how it affects the types of problems/applications that can be considered by this work?

The authors have not provided any justification for their choice of reward on page 4. If the black-box function is indeed complex and highly varying, the distance between points may not work well at all. Can the authors provide a justification and discuss the practical implications and limitations with such a choice?

Isn't it more natural to consider a single Bayesian neural network instead of using a neural network for the mean and a GP for the variance?

For the experiments, it would be good to see two other variants of the proposed algorithm to understand the individual contributions of the neural surrogate model and weighted Reptile algorithm: one without neural surrogate model and the other with simply the use of neural surrogate model.

Can the authors explain in greater detail how they run their algorithms (Algorithms 2 and 3) in the experiments? For example, the authors say that "WRA-N starts with learned initial surrogate model". I assume that WRA-N refers to Algorithm 3 based on its acronym. Isn't the learned initial surrogate model the output of WRA-N in the first place? Also, the graphs in Fig. 2 seem to show iteration 1 to 13 in NOE (Algorithm 2). However, Algorithm 3 accepts N_T = 13 and executes NOE for N_T = 13 (and not 1, 2, or 3, ...) for each problem in each epoch. How do the authors generate the plot of WRA-N for iterations 1 to 12?
What seems to make more sense to me is that the authors in fact run Algorithm 2 instead of Algorithm 3 for each experiment and they initialize w in Algorithm 2 to the output of Algorithm 3. In any case, a clarification is needed here.

It is not clear to me whether the initial/prior information from related problems/meta tasks provided to WRA-N, TST-R, AND TSR-M is fair. Can the authors provide a justification?

To clarify, for each related problem/function, only N_T number of datapoints are used to train a corresponding neural network with 1 hidden layer of 15 hidden units?

The authors say that "Since TST-R needs base models for combination, we sample 20 points from uniform distribution in (−10, 10) to construct base models." Is this sampling procedure the same as that in (Wistuba et al., 2016)?

Can the authors explain the comparable performance of WRA-N and TST-R in Fig. 5? Why are the error bars missing?

How does the proposed approach compare with that of Feurer et al. (2018)?



Minor issues

Page 1, 3: adapt well to new tasks.
Page 2: The author says "depends on a GP-based surrogate model fitting function values without learnable parameters". This is not true: The GP hyperparameters need to be learned and they adapt to new problems.
Page 4: descent order?
Pages 4, 5: Why is there an input x to Q_i?
Algorithm 2: t^* should be at the superscript of x.
Equation 7: What is N?
Page 5: Does it make a difference in the performance when delta is set to 0?
Page 5: well define meta-features?

**Experience Assessment:**

I have published in this field for several years.

**Review Assessment: Checking Correctness Of Derivations And Theory:**

I carefully checked the derivations and theory.

**Review Assessment: Checking Correctness Of Experiments:**

I carefully checked the experiments.

**Review Assessment: Thoroughness In Paper Reading:**

I read the paper thoroughly.

---

> ### Author Response · Authors · 2019-11-15
> **Rebuttal**
>
> Thanks for your review,
> 1 Feurer et al. (2018) used linear combination of variance of related problems. It is not reasonable,  first as the variance of  related problems is not determinate as it depends on sampling points. Thus different sampling points can cause different variance, then cause different surface in new problem. Second, in related problems, lower variance means lower uncertainty, then should this lower uncertainty bring to new problem?  If so, then in new problems, the variance of points sampled in related problems are low and the variance of points not sampled in related problems are high. It is absurd.
> 2 Knowing accurate global minimizer is acctually hard, however, we can use global minimizer of surrogate model. In other words, when meeting a problem, we sampled some points on it and construct a surrrogate model, then using this global minimizer to substitute.
> 3 Actually, when tuning hyper-parameters for neural networks, we may have tuned several similar (in structure )networks. Then these knowledge can be used to new tuning process.
> 4 May be global minimizer is not a reasonable setting. However, using function value is more unreasonable. Our aim is to finding global minimizer of objective function, thus using distance of global minimizer seems no problem.
> 5  'For the experiments, it would be good to see two other variants of the proposed algorithm to understand the individual contributions of the neural surrogate model and weighted Reptile algorithm: one without neural surrogate model and the other with simply the use of neural surrogate model.'
> I can give you the comparation about initialization on ablation study (on problem of synthetic function):
> initialization    |2.1|1.2|  0  |-0.8|-1.1|-1.2|-1.3|-1.4|-1.4|-1.4
> after training  |0.1|  0 |-0.7|-1.2|-2.4|-3.5|-4.1|-4.8|-5.0|-5.9
> But how to use WRA without neural surrogate model? WRA is a training method to train neural surroagte models. Without  neural surrogate model, what should WRA learn?
> 6 'What seems to make more sense to me is that the authors in fact run Algorithm 2 instead of Algorithm 3 for each experiment and they initialize w in Algorithm 2 to the output of Algorithm 3.' You are right, we do not write clearly. We actually use Alg 2 to get  iterations 1 to 12 and its initialization is got by Alg 3.
> 7 'To clarify, for each related problem/function, only N_T number of datapoints are used to train a corresponding neural network with 1 hidden layer of 15 hidden units?' We actually use neural network with 1 hidden layer of 15 hidden units.
> 8   'sample 20 points from uniform distribution in (−10, 10)' is the same as Feurer et al. (2018) who has compared TST-R.
> 9 TST-R based on linear combination of related problems, thus related problems determine the performance of TST-R. The comparable performance in SVM problem may beacuse this data set is suitable to TST-R. However, other datasets are not so suitable.
> 10  I can give you the comparation about Feurer et al. (2018) on ablation study (on problem of synthetic function):
> Feurer et al. (2018)   |0.1|  0 |-1.7|-1.7|-1.7|-1.7|-1.7|-2.0|-2.0|-2.1
> WRA-N                        |0.1|  0 |-1.9|-2.7|-2.8|-2.8|-3.5|-4.1|-4.4|-5.0

---

### Decision · Program_Chairs · 2019-12-19

**Decision:**

Reject

**Comment:**

This paper is concerned with warm-starting Bayesian optimization (i.e. starting with a better surrogate model) through transfer learning among related problems.

While the key motivation for warm-starting BO is certainly important (although not novel), there are important shortcomings in the way the method is developed and demonstrated. Firstly, the reviewers questioned design decisions, such as why combine NNs and GPs in this particular way or why the posterior variance of the hybrid model is not calculated. Moreover, there are issues with the experimental methodology that do not allow extraction of confident conclusions (e.g. repeating the experiments for different initial points is highly desirable). Finally, there are presentation issues. The authors replied only to some of these concerns, but ultimately the shortcomings seem to persist and hint towards a paper that needs more work.